# Fibroblast Growth Factor 21 Response in a Preclinical Alcohol Model of Acute-on-Chronic Liver Injury

**DOI:** 10.3390/ijms22157898

**Published:** 2021-07-23

**Authors:** Grigorios Christidis, Ersin Karatayli, Rabea A. Hall, Susanne N. Weber, Matthias C. Reichert, Mathias Hohl, Sen Qiao, Ulrich Boehm, Dieter Lütjohann, Frank Lammert, Senem Ceren Karatayli

**Affiliations:** 1Department of Medicine II, Saarland University Medical Center, Saarland University, 66421 Homburg, Germany; Grigorios.Christidis@uks.eu (G.C.); Ersin.Karatayli@uks.eu (E.K.); Rabea.Hall@uks.eu (R.A.H.); Susanne.Weber@uks.eu (S.N.W.); Matthias.Reichert@uks.eu (M.C.R.); Frank.Lammert@uks.eu (F.L.); 2Department of Medicine III, Saarland University Medical Center, Saarland University, 66421 Homburg, Germany; Mathias.Hohl@uks.eu; 3Department of Pharmacology and Toxicology, Saarland University Medical Center, Saarland University, 66421 Homburg, Germany; Sen.Qiao@uks.eu (S.Q.); Ulrich.Boehm@uks.eu (U.B.); 4Institute of Clinical Chemistry and Clinical Pharmacology, University Hospital Bonn, 53127 Bonn, Germany; Dieter.Luetjohann@ukbonn.de; 5Hannover Health Sciences Campus, Hannover Medical School (MHH), 30625 Hannover, Germany

**Keywords:** acute-on-chronic liver failure (ACLF), alcohol-associated liver disease (AALD), ATP binding cassette subfamily B member 4 (*Abcb4*) knock-out mouse, bile acid, cholesterol 7α-hydroxylase (Cyp7A1), fibroblast growth factor 21 (FGF21)

## Abstract

Background and Aims: Fibroblast growth factor (FGF) 21 has recently been shown to play a potential role in bile acid metabolism. We aimed to investigate the FGF21 response in an ethanol-induced acute-on-chronic liver injury (ACLI) model in *Abcb4^−/−^* mice with deficiency of the hepatobiliary phospholipid transporter. Methods: Total RNA was extracted from wild-type (WT, C57BL/6J) and *Abcb4^−^/^−^* (KO) mice, which were either fed a control diet (WT-Cont and KO-Cont groups; *n* = 28/group) or ethanol diet, followed by an acute ethanol binge (WT-EtOH and KO-EtOH groups; *n* = 28/group). A total of 58 human subjects were recruited into the study, including patients with alcohol-associated liver disease (AALD; *n* = 31) and healthy controls (*n* = 27). The hepatic and ileal expressions of genes involved in bile acid metabolism, plasma FGF levels, and bile acid and its precursors 7α- and 27-hydroxycholesterol (7α- and 27-OHC) concentrations were determined. Primary mouse hepatocytes were isolated for cell culture experiments. Results: Alcohol feeding significantly induced plasma FGF21 and decreased hepatic *Cyp7a1* levels. Hepatic expression levels of Fibroblast growth factor receptor 1 (*Fgfr1*), *Fgfr4*, Farnesoid X-activated receptor (*Fxr*), and Small heterodimer partner (*Shp*) and plasma FGF15/FGF19 levels did not differ with alcohol challenge. Exogenous FGF21 treatment suppressed *Cyp7a1* in a dose-dependent manner in vitro. AALD patients showed markedly higher FGF21 and lower 7α-OHC plasma levels while FGF19 did not differ. Conclusions: The simultaneous upregulation of FGF21 and downregulation of *Cyp7a1* expressions upon chronic plus binge alcohol feeding together with the invariant plasma FGF15 and hepatic *Shp* and *Fxr* levels suggest the presence of a direct regulatory mechanism of FGF21 on bile acid homeostasis through inhibition of CYP7A1 by an FGF15-independent pathway in this ACLI model. Lay Summary: Alcohol challenge results in the upregulation of FGF21 and repression of *Cyp7a1* expressions while circulating FGF15 and hepatic *Shp* and *Fxr* levels remain constant both in healthy and pre-injured livers, suggesting the presence of an alternative FGF15-independent regulatory mechanism of FGF21 on bile acid homeostasis through the inhibition of Cyp7a1.

## 1. Introduction

Excessive alcohol consumption is associated with serious adverse effects in the liver, causing alcohol-associated liver disease (AALD) with a wide pathobiological and histopathological spectrum ranging from fat accumulation in hepatocytes to hepatocellular carcinoma (HCC) [1]. In the setting of such chronic liver diseases, an acute hepatic insult may have devastating effects on liver pathophysiology as in the case of acute-on-chronic liver injury (ACLI) [2] or acute-on-chronic liver failure (ACLF), which leads to multi-organ and -system failure, and high short-term mortality [3]. Various animal models have been established so far to mimic human AALD, among which murine models are the most preferred ones in experiments for a variety of reasons, including the ease of maintenance, cost effectiveness, as well as short reproductive cycle and lifespan [4,5]. Murine models fed ad libitum the Lieber-DeCarli (LDC) liquid diet [6], which is the most commonly used alcohol-containing liquid diet, presented with elevated serum aminotransferase activities and hepatic steatosis with mild inflammation but no fibrosis [7]. However, with the modification of the LDC diet model such as chronic-plus-binge alcohol feeding, i.e., the NIAAA model of the National Institute on Alcohol Abuse and Alcoholism (NIAAA) (Gao-binge model), higher aminotransferase levels, liver inflammation and steatosis, and even hepatic neutrophil infiltration mimicking the human condition could be established [8,9].

For better understanding of alcohol-induced acute inflammatory responses in chronically injured liver, we recently modeled alcohol-associated ACLI based on the chronic plus binge ethanol feeding model [8] in ATP binding cassette subfamily B member 4 knock out (*Abcb4^−/−^*) mice, which resulted in exacerbated hepatic steatosis, liver injury, and inflammation [2]. *Abcb4* encodes the ATP-dependent phosphatidylcholine (PC) translocating protein (ABCB4) that flops PC from the inner to the outer leaflet of the hepatocanalicular membrane. ABCB4 deficiency induces chronic liver inflammation and progressive hepatic fibrosis in *Abcb4^−^/^−^* mice [10,11]. In our novel preclinical ACLI mouse model that mimics ACLF, exacerbated hepatic steatosis, liver injury, and inflammation have been documented, which serves to dissect liver-specific pathomechanisms [2]. One of these mechanisms in liver includes the response of fibroblast growth factor 21 (FGF21), a member of the endocrine subfamily of FGFs with a pivotal role in the regulation of metabolism. FGF21 has previously been suggested as a predictor for diagnosis and prognosis of ACLF in critically ill patients [12].

Several studies have suggested FGF21 as an emerging regulator of the preference for alcohol in rodents and humans, which shows markedly increased expression levels following acute and chronic alcohol consumption [13,14,15,16,17]. This induction of FGF21 expression in the liver, where it is mainly expressed, was proposed to be hepatoprotective when liver homeostasis is challenged by various stimuli, as in the case of alcohol [15,18,19,20]. Recently, a previously unidentified additional role of FGF21 as a negative regulator of bile acid synthesis has been described [21,22].

Bile acids are essential to facilitate digestion and absorption of dietary fats, steroids, and lipid-soluble vitamins in the small intestine and act as signaling molecules for lipid and glucose metabolism in the liver [23,24,25]. Several studies have indicated altered hepatic bile acid synthesis and accumulation as a potential major pathophysiological trigger for the induction of the structural and functional hepatocellular changes in both alcohol-associated steatohepatitis and metabolic dysfunction-associated fatty liver disease [26,27,28]. Thus, the bile acid feedback loop, the molecular mechanisms of which have been studied for many years, plays an essential role in protecting against liver damage. Among several cytochrome P450 (CYP) hydroxylase enzymes involved in bile acid synthesis, cholesterol 7α-hydroxylase (CYP7A1), encoded by the cholesterol 7α-hydroxylase (*CYP7A1*) gene, is the critical rate-limiting enzyme in the classic bile acid synthesis pathway for the conversion of cholesterol into 7α-OHC [29]. Although controversial results have been shown for the effects of alcohol on Cyp7a1 gene expression levels [26,30], alcohol-induced inhibition of Cyp7a1 expression has been recently suggested to contribute to the protection against alcohol-associated steatohepatitis [31]. Despite the evidence of a hepatoprotective role of both FGF21 and CYP7A1 in alcohol-associated liver disease, the interacting mechanism of these two key regulators is a new area that remains to be clarified not only during alcohol consumption itself but also in the frame of different alcohol drinking patterns.

The primary mechanisms suggested for the inhibition of bile acid synthesis are the induction of intestine farnesoid X receptor/fibroblast growth factor 19 (FXR/FGF19) and the activation of the farnesoid X receptor/small heterodimer partner (FXR/SHP) axis in liver by elevated bile acid levels, which results in the inhibition of *Cyp7a1* transcription [32,33]. This repression is due to elevated bile acid concentrations in the ileum activating FXR, which induces the secretion of the enterokine FGF19 (FGF19; FGF15 in mice) into the portal circulation and the consequential stimulation of fibroblast growth factor receptor 4 (FGFR4)/β-Klotho (KLB) complex in the hepatocyte membrane [32,34]. As though it was initially hypothesized that FGF21 antagonizes this action by inhibiting the FGF15/19-mediated suppression of *Cyp7a1* expression [22], a recent study indicated that FGF21 has a novel role in bile acid metabolism, acting as a negative regulator of bile acid synthesis independent of the FXR/FGF15 pathway [21].

Therefore, in view of these initial data and accumulating evidence, we aimed to investigate the impact of chronic plus binge alcohol-induced FGF21 on bile acid metabolism in our recently established ACLI mouse model with pre-existing chronic cholestatic liver injury. For the clinical validation of our findings from animal studies, we also examined the effect of alcohol intake on FGF19, FGF21, and bile acid metabolism in patients with alcohol-associated liver cirrhosis and healthy subjects as a control group. To better understand the bile acid mechanism in our ACLI model, bile acid compositions were evaluated in both murine and human plasma samples. Bile acid compositions were determined in both murine and human plasma samples to better understand the bile acid metabolism in our ACLI model.

## 2. Results

### 2.1. ACLI Model

#### 2.1.1. Effect of Ethanol Consumption on FGF21

Ethanol exposure significantly upregulated the hepatic *Fgf21* gene expression in both WT and KO mice (*p* = 0.009 and 0.011, respectively), as compared to their normal diet-fed controls, showing no difference between the two genotypes (Figure 1A). Hepatic protein levels confirmed the upregulation of FGF21 in randomly chosen EtOH-challenged mice (Figure 1B). Consequently, a significant elevation of FGF21 concentrations in plasma samples of EtOH-challenged WT and KO mice (*p* = 0.040 and 0.048, respectively) was ascertained, consistent with the existing literature reporting that hepatic expression confers the major part of circulating FGF21 (Figure 1C) [35]. No difference in relative FGF21 expression in the WT-EtOH group was observed as compared to KO-EtOH mice (WT-EtOH/WT-Cont compared to KO-EtOH/KO-Cont).

To investigate the effect of alcohol on FGF21 signaling pathways, hepatic mRNA levels of the *Ppar-a* (FGF21 upstream mediator PPARα [36]), *Mtor* (downstream mediator mTOR [37]), and *Srebf1/Srebp1c* genes (transcription regulator SREBP1) were evaluated. Hepatic *Ppar-a* expression levels showed significant upregulation after the EtOH challenge in both WT and KO mice (*p* = 0.01 and 0.02, respectively), with no difference between genotypes (Figure 2A). No effects of alcohol challenge on hepatic mRNA levels of *Fgfr1*, which represents the preferred receptor of FGF21, were observed, while the down-regulation of its obligate co-receptor β-klotho (*Klb*) did not reach statistical significance (Figure 2B,C). In addition, no differences were observed for hepatic mRNA expression of *Srebf1/Srebp1c* and *Mtor* between groups (Figure 2D,E). Western blots to investigate mRNA/protein correlations and the activity of mTOR after post-translational modifications in mouse livers, further confirmed the relative hepatic mRNA quantification results (Figure 2F), with no statistically significant differences between groups in ratios of *p*-mTOR (Ser2448) to total mTOR. These results confirm the effect of alcohol on FGF21 upregulation and also suggest that PPARα, but not the transcription factor SREBP1, may have a direct impact on FGF21 upregulation. Moreover, our results indicate that the downstream signal transduction of FGF21 is possibly not mediated through the mTOR pathway, which was previously reported to play a potential role in the FGF21 signaling network [37].

#### 2.1.2. Relative Expression Levels of Bile Acid Synthesis Genes and Plasma and Gallbladder Total Bile Acid Levels

The hepatic expression of *Cyp7a1* and *Cyp27a1*, which encodes sterol 27-hydroxylase (CYP27A1) in the alternative BA pathway, was significantly repressed in WT-EtOH (*p* < 0.001 and 0.012, respectively) and KO-EtOH mice (*p* = 0.014 and 0.001, respectively), as compared to WT-Cont mice (Figure 3A,C). Hepatic protein levels of CYP7A1 in randomly chosen mice confirmed the repression in EtOH-challenged mice (Figure 3B). Significant repression of the *Cyp8b1* gene, encoding sterol 12α-hydroxylase (CYP8B1), which is required for synthesis of cholic acid, was observed in KO-EtOH mice only, when compared to the WT-Cont (*p* = 0.041), WT-EtOH (*p* = 0.029), and KO-Cont groups (*p* = 0.027) (Figure 3D).

#### 2.1.3. Hepatic FXR/SHP Pathway

To understand the molecular pathway of *Cyp7a1* repression under EtOH challenge, the hepatic mRNA and protein expression levels of nuclear receptor FXR, which inhibits *Cyp7a1* and *Cyp8b1* gene transcriptions by inducing the nuclear receptor SHP, were analyzed [33,38]. However, no significant difference in expression levels of hepatic *Fxr* and *Shp* was observed between groups (Figure 4A,B, Appendix A), suggesting that the repression of *Cyp7a1* gene expression was independent of the FXR/SHP pathway in our ACLI model.

#### 2.1.4. FXR/FGF15/FGFR4 Pathway

Another mechanism of *Cyp7a1* repression is known to be based on the activation of FXR in the ileum, which induces the intestinal hormone FGF15 that activates hepatic FGFR4 signaling [34]. The significant upregulations of ileal *Fgf15* (*p* = 0.032) and *Fxr* gene expressions (*p* = 0.018) in WT mice were observed upon EtOH uptake but not in the KO group (Figure 4D,E). Interestingly, the circulating total FGF15 plasma levels did not differ between groups (Figure 5). In mice, ileal FGF15 is secreted into the portal circulation and carried to the liver, where it binds to its cognate receptor FGFR4 and is then released into the systemic circulation. Therefore, higher levels of FGF15 might be expected in portal as compared to systemic blood. However, although one should be cautious about plasma FGF15 levels reflecting the portal concentrations, using targeted mass spectroscopy, Katafuchi et al. showed a strong correlation between ileal and plasma FGF15 levels [39], suggesting a correlation with portal FGF15 as well. To clarify this result, we also measured the ileal expression level of the *Diet1/Malrd1* gene, which has been suggested to be related to the levels of FGF15 secretion into the portal circulation [40]. Although ileal *Diet1* expression paralleled the expression pattern of *Fgf15* and *Fxr*, with higher levels in WT mice challenged with EtOH, no differences between groups were observed for *Diet1* (Figure 4F) and hepatic *Fgfr4* (Figure 4C). Moreover, no statistical difference for the co-receptor *Klb* expression was observed between groups, albeit its expression was lower in the EtOH group (Figure 2B). The lack of significant changes in the ileal expression levels of *Fxr* and *Fgf15*, as well as plasma FGF15 levels in the KO-EtOH group, suggests that the alcohol-dependent upregulation of FGF21 may have a direct link to *Cyp7a1* repression in the ACLI model.

#### 2.1.5. Total Bile Acid Concentrations and Composition in Mice

In order to evaluate the changes in bile acid metabolism among the experimental groups, TBA concentrations and composition were determined. Although statistically not significant, chronic plus binge ethanol feeding increased TBA concentrations both in the plasma and gallbladder bile of mice (Figure 6A,B), showing higher plasma TBA levels in KO-Cont mice as compared to the WT-Cont group. These findings indicate that although the expression of genes involved in the bile acid synthesis cascade was repressed (Figure 3), the bile acid pool size was increased in murine plasma and gallbladders after ethanol feeding. A lower percentage of secondary bile acids (DCA, LCA, and UDCA) was observed in KO-EtOH (4.1%), and the highest percentage was in WT controls (16.6%) (Figure 6C and Table 1). The most abundant bile acids in plasma were found to be CA and βMCA in all groups. In the WT-Cont group, the pool contained 26.3% CA and 49.1% βMCA (Figure 6C and Table 1). Compared to this group, WT-EtOH and KO-EtOH groups showed reduced βMCA (42.1% and 36.9%, respectively). The KO-Cont group presented higher CA (46.2% vs. 26.3%) and lower βMCA (36.9% vs. 48.5%) levels (Figure 6C and Table 1). High βMCA (62.7%) with low CA (25.9%) and CDCA (1.0%) levels suggest a shift to more hydrophilic BA with the repression of hepatic *Cyp8b1* and *Cyp7a1* in EtOH-treated KO mice.

#### 2.1.6. Effects of FGF21 on Cyp7a1 Expression Levels in Primary Mouse Hepatocytes

In mouse primary hepatocytes, rhFGF21 decreased *Cyp7a1* expression levels in a dose-dependent manner. Although the expression level of *Cyp7a1* was slightly decreased with lower FGF21 doses (0.1 and 0.25 µg/mL), significant suppression was observed at 0.5 and 1 µg/mL doses by 25% and 70% relative to the control, respectively (Figure 7A). In order to assess whether these changes were mediated by a direct or indirect action of FGF21, expression levels of *Fxr* and *Shp* were evaluated, and no difference was observed for either of the two genes (Figure 7B,C). Additionally, no changes in gene expression of hepatic *Fgf21*, *Fgfr1*, and *Klb* were observed after the administration of exogenous FGF21 (Figure 7D,F). In line with our in vivo findings, these results suggest a direct role of exogenously administered FGF21 on *Cyp7a1* gene suppression, which is independent of the FXR/SHP pathway.

#### 2.1.7. Plasma ALT, AST and Ethanol Levels in Mice

In order to validate the impact of chronic plus binge ethanol consumption on liver function, we compared serum aminotransferase (ALT and AST) activities, as well as EtOH levels, before and after the alcohol challenge in mouse groups. As expected, high levels of EtOH were detected in plasma samples of WT and KO mice challenged with EtOH (Appendix A). The serum aminotransferases were significantly elevated in both WT and KO mice as compared to their control diet-fed controls (*p* < 0.001 for ALT and *p* = 0.013 for AST in WT mice, *p* = 0.001 for ALT and *p* = 0.001 for AST in KO mice) (Appendix A), indicating aggravated liver injury.

#### 2.1.8. Histological Assessment of Murine Liver Injury

Increased numbers of infiltrating inflammatory cells shown by H&E staining indicated increased hepatic steatosis in mice challenged with EtOH regardless of genotype (Appendix A). Sirius red staining of liver sections showed that chronic plus binge ethanol feeding aggravated liver fibrosis in both WT and KO mice (Appendix A). In mice challenged with ethanol, Hyp contents and collagen area calculation indicated higher collagen levels; however, this increase was not significant when compared to the control diet-fed counterparts (Appendix A). The inflammatory cell infiltrates were composed of F4/80^+^ macrophages, CD45^+^ neutrophils, and CD11^+^ and CD4^+^ lymphocytes (Appendix A).

### 2.2. Human Studies

In order to translate our findings to humans, we evaluated the effects of EtOH consumption on FGF21 and bile acid metabolism, by analyzing FGF21, FGF19 (the human orthologue of FGF15 in mice), 7α-hydroxycholesterol (7α-OHC), and oxysterol 27-hydroxycholesterol (27-OHC) levels. Healthy controls did not have any known medical history and presented with normal BMI (18–25 kg/m^2^). For the evaluation of alcohol intake and liver injury, biochemical parameters were measured in serum samples of both groups. As expected, CDT, ALT, and AST levels were significantly elevated in AALD patients with ongoing alcohol intake (*p* < 0.001, *p* = 0.004, and *p* < 0.001, respectively) (Appendix A).

In parallel to the observed results in mice, a significant (*p* < 0.001) upregulation of FGF21 in the cohort with ongoing EtOH consumption was found, suggesting that the previously ascertained induction of FGF21 by EtOH consumption is preserved in AALD patients (Figure 8A). The circulating plasma FGF19 levels in humans, which has been reported to reflect portal concentrations in previous studies [41,42], remained unchanged between the groups, which is in line with the measurements of plasma FGF15 concentrations in mice (Figure 8B).

To evaluate the potential changes in *CYP7A1* and *CYP27A1* genes in humans, we evaluated the serum concentrations of the specific metabolite 7α-OHC, which has been proposed as an indicator of 7α-hydroxylase activity [43], and 27-OHC levels in human serum samples, respectively. Consistent with the findings in mice, a significant (*p* = 0.02) suppression of 7α-OHC was observed in AALD patients (Figure 9A), which appears to be mostly due to significant suppression in female AALD patients (Figure 9B). However, levels of 27-OHC did not differ between groups (Figure 9C).

In humans, serum TBA concentrations were found to be significantly (*p* = 0.01) elevated despite a nonsignificant reduction of total cholesterol levels in the AALD cohort as compared to controls (Figure 10A,B). Altogether, this data suggests a significant upregulation in TBA and FGF21 levels in the plasma of AALD patients in the absence of marked changes in FGF19 levels upon EtOH consumption. To evaluate the bile acid pool composition in healthy controls and AALD patients, serum bile acid profiles were determined by GC-MS. Overall, a greater percentage of the secondary bile acid DCA was observed in healthy controls (Figure 10D and Table 2) as compared to AALD patients. As expected, the total percentage of secondary bile acids was found to be markedly lower in patients with AALD (27.8%) than in healthy subjects (50.3%). Accordingly, increased levels of CDCA and CA were observed in patients with AALD as compared to healthy subjects.

## 3. Discussion

The FGF21 response and its relation to hepatic bile acid homeostasis are described for the first time in the present study in an alcohol-induced acute-on-chronic liver injury model, which has previously been described by our group [2]. The study identifies a significant induction of hepatic and serum FGF21 levels in both human AALD patients and rodents with pre-injured liver upon EtOH consumption, supporting that this reaction is preserved even in the context of pre-existing hepatic pathology [12,44]. Furthermore, in vivo studies have indicated a significant downregulation of hepatic *Cyp7a1* expression after chronic plus binge ethanol feeding in mice, regardless of the presence of pre-existing liver injury. The results of our in vitro studies suggest a direct and dose-dependent effect of FGF21 on hepatic *Cyp7a1* mRNA suppression. Moreover, due to the lack of any changes in expression of hepatic *Fxr* and *Shp* or plasma FGF15/19 levels after EtOH challenge, we propose that alcohol-induced FGF21 has a direct effect on *Cyp7a1* regulation and thus on bile acid homeostasis, which is independent of the canonic FXR-SHP or FGF15/FGFR4 pathways (Figure 11). In accordance with the in vivo results, our translational human studies demonstrate significant FGF21 upregulation and reduction of serum 7αOHC levels in AALD patients upon ongoing EtOH consumption together with unchanged plasma FGF19 levels.

The direct role of FGF21 in the regulation of bile acid synthesis and its interaction with FGF15/19 is a controversial issue. In a recent study by Chen et al. [21], in which rhFGF21 and its long-acting analogs, as well as rhFGF19, were administered in different rodent models at pharmacologic doses, FGF21 has been proposed for the first time as a negative regulator of bile acid metabolism in a manner that is independent of the FGF15/19 pathway. In the experimental study, mice treated with rhFGF19 showed decreased gene expression in bile acid synthesis pathways. However, acute or long-term administration of FGF21 showed different results on *Cyp7a1* gene expression. Although the acute administration of FGF21 dose-dependently inhibited the hepatic expression of *Cyp7a1* in inbred mice and human primary hepatocytes, which is in line with the in vivo and in vitro findings of the present study, long-term administration resulted in the inhibition of *Cyp7a1* expression only in mice treated with long-acting FGF21, but not in mice treated with rhFGF21. Moreover, despite the reduced hepatic *Cyp7a1* expression at lower doses of rhFGF21 (0.3 mg/kg), mice treated with higher rhFGF21 doses (3 mg/kg) demonstrated elevated *Cyp7a1* levels, suggesting a potential feedback mechanism between FGF21 and *Cyp7a1* expression depending on dose and exposure [21]. In another recent study, in which a different mechanism of action for FGF21 on bile acid metabolism was suggested, Zhang et al. [22] proposed that the overexpression of FGF21 regulates bile acid homeostasis by antagonizing FGF15/19 function on the liver βKlotho/FGFR4 receptor complex, resulting in the inhibition of FGF15-mediated *Cyp7a1* downregulation. The study showed the induction of *Cyp7a1* and *Cyp27a1* expression through FGF21 overexpression. Interestingly, in both studies, the upregulation of *Cyp7a1* levels was observed with the long-term administration of high doses of FGF21. Indeed, the method of FGF21 overexpression that was used by Zhang et al. [22] was through delivery of the *Fgf21* gene by a recombinant adeno-associated virus (AAV) approach that might result in higher doses of FGF21, as well as longer-acting effects, as compared to alcohol-induced FGF21 effects. Despite the induced expression of *Cyp7a1* through chronic FGF21 treatment, the results of the acute administration showed a decrease in *Cyp7a1* levels at 2 and 5 h post-injection, without any difference in *Fgf15* expression [22], pointing to a potential feedback mechanism between FGF21 and *Cyp7a1* to maintain bile acid homeostasis, which might depend on the level and duration of action of FGF21. FGF21 itself has been shown to be hepatoprotective previously [15,18,19,20]. This hepatoprotective role may be carried out by maintaining bile acid homeostasis through regulation of *Cyp7a1* by FGF21, as CYP7A1 is the rate-limiting enzyme in the classical pathway of bile acid synthesis. In this regard, a study performed by Donepudi et al. [31] reported alcohol-induced downregulation of *Cyp7a1* and a hepatoprotective role in *Cyp7a1* transgenic mice. Similar to the current study, mice were challenged with the NIAAA chronic plus binge ethanol feeding method [8]. Although FGF21 was not evaluated, a marked suppression of hepatic *Cyp7a1* levels in alcohol-challenged mice compared with their pair-fed controls were reported, together with aggravated liver injury and inflammation, as well as increased bile acid pool size. These results were in line with ours except for the gender-dependent finding that *Cyp7a1* downregulation was observed in female mice only, whereas the expression was not altered in male mice. In our model, a marked downregulation of hepatic *Cyp7a1* was observed after EtOH challenge in both female and male mice, but, interestingly, 7αOHC was significantly suppressed only in female AALD patients with ongoing EtOH abuse (Figure 9B). Taking the possible relevant and direct role of sex hormones on bile acid metabolism into account [45], these differences between genders might depend on the fluctuations of female sex hormone levels.

In addition to alcohol, the induction of FGF21 expression has also been reported in response to glucocorticoids [46]. Of note, instead of a distinct trigger, the study proposed an effector mechanism of FGF21, in which a counterbalanced mechanism, including *Cyp7a1* repression through upregulated FGF21, occurs in a PPARα-dependent manner, irrespective of the FXR/FGF15 pathway. The study also reported that the in vitro downregulation of *Cyp7a1* occurs by the inhibition of the transcriptional activity through an FGF21-mediated autocrine signal, which might explain the direct effect of FGF21 on the suppression of *Cyp7a1* expression observed in our in vitro experiment. Besides, the suggested mechanism also involves the upregulation of PPARα and IL-6 in line with the findings in our model [2], pointing to a similar mechanism of action for FGF21 even with different acute stimuli such as steroids and alcohol. In order to investigate the effect of alcohol on FGF21 downstream factors, the mTORC1 pathway, which promotes hepatic lipogenesis by activating SREBP1 [47], was evaluated. However, no difference between the study groups was observed for expression levels of the *Mtor* and *Srebf1/Srebp1c* genes. Moreover, the ratio of phosphorylated to total mTOR protein, which points to the activity [48], did not differ between groups, indicating that this downstream pathway is not affected by alcohol-induced FGF21 signaling (Figure 2F).

Higher plasma and gallbladder bile acid levels were observed in mice with alcoholic liver disease as compared to their controls. Although this upregulation was neither significant in plasma nor in the gallbladder, an increased TBA concentration points to alcohol-induced intrahepatic cholestasis. Moreover, higher plasma TBA levels in KO-Cont mice in comparison to WT-Cont mice are consistent with the phenotypes of *Abcb4^−^/^−^* mice, which develop cholestasis due to the deficiency of hepatobiliary phosphatidylcholine transport [11]. Increased bile acid levels are observed despite the repression of hepatic genes involved in bile acid synthesis. Similarly, Donepudi et al. [24] reported that chronic plus binge ethanol feeding in mice resulted in increased intestinal bile acid contents and total bile acid pooled together with suppressed bile acid synthesis genes. Although the underlying mechanisms are yet to be dissected, changes in the enterohepatic circulation of bile acids upon alcohol exposure might play a role, as alcohol represses bile acid uptake by sodium taurocholate cotransporting peptide (NTCP) while inducing bile acid efflux systems such as the bile salt export pump (ABCB11) [27,28,49]. Similarly, pre-established, EtOH-induced cholestatic liver injury with elevated serum TBA levels has been reported previously in patients with AALD and ongoing alcohol consumption as compared to healthy subjects [50,51]. Total cholesterol serum levels were slightly reduced, while a mild elevation of triglyceride levels was observed, which is an expected pattern considering the impact of EtOH on lipid metabolism [52].

Although the hepatic steady-state mRNA levels of the bile acids synthesis enzymes do not completely reflect their activities, we evaluated the hepatic expression of *Cyp7a1*, *Cyp8b1*, and *Cyp27a1*. Among these genes, the downregulation of *Cyp7a1* and *Cyp27a1* was found to be alcohol-dependent irrespective of the mouse genotype. However, a significant suppression of *Cyp8b1* upon EtOH challenge was observed in *Abcb4^−^/^−^* knock-out mice only, emphasizing the importance of established chronic liver injury in the presence of acute insult, as it is the case in our ACLI model. In a recent study, Wang et al. [53] reported seven genes that may serve as potential prognostic biomarkers for HCC, showing a significant downregulation as compared to normal human liver tissue. Among these genes, *CYP8B1* was suggested to be involved in tumor initiation and development. Therefore, the downregulation of *Cyp8b1* in the ACLI model might trigger harmful consequences during liver disease progression, further increasing the known risk of HCC in *Abcb4^−^/^−^* mice [54]. Similar to *Cyp8b1*, the observed decrease in expression of hepatic *Klb* only in the KO-EtOH group, might suggest the modulating effect of pre-injured liver in ACLI. Despite the studies supporting its role in tumorigenesis [55,56], its exact role in ACLI needs further investigation.

Bile acid composition differs in various chronic liver diseases [57]. To better understand the mechanisms that regulate bile acid metabolism, we analyzed the bile acid composition in both human and mouse plasma samples. In the current preclinical ACLI mouse model, the bile acid pool showed higher MCA and lower CA and CDCA contents, consistent with a shift toward the alternative bile acid synthesis pathway. Indeed, in this model, a significant repression of hepatic *Cyp8b1* and *Cyp7a1* levels was observed, although the *Cyp27a1* expression was also reduced, but this enzyme is expressed in various tissues and is not rate-limiting. *Cyp8b1* is negatively regulated by bile acids [58], with CDCA inducing the negative nuclear receptor SHP [59,60] and MCA representing an antagonist of the central bile acid sensor FXR [61]. However, in our study, *Shp* and *Fxr* gene expression did not differ in the KO-EtOH group, and other feedback mechanisms have yet to be determined. Although there are several differences between humans and mice regarding BA metabolism [62], in the current study, AALD patients showed increased bile acid concentrations and decreased secondary bile acids levels, which is in line with the ACLI mouse model. The decreased levels of hydrophobic bile acids are consistent with recent findings by Sugita et al. [57] and may reflect the impairment of bile excretion.

In summary, the current study suggests that chronic plus binge alcohol-induced FGF21 suppresses the gene encoding the bile acid synthesis rate-limiting enzyme *Cyp7a1*, independent of the FXR-SHP or FGF15/FGFR4 pathways. Although the interaction between FGF21 and CYP7A1 in specific settings remain to be defined, the observed upregulation of FGF21 and suppression of *Cyp7a1* were preserved in the context of pre-existing liver injury. The role of these differential expression levels will be the subject of future research.

## 4. Materials and Methods

### 4.1. Mouse Model

Fifteen-week-old C57BL/6J wild-type (WT, *n* = 56) and *Abcb4* knock-out (KO) mice (*Abcb4^−/−^*, *n* = 56) housed in a temperature- and humidity-controlled facility with a 12 h light and 12 h dark cycle were included in the study. Mice were either fed a control diet (WT-Cont and KO-Cont groups; *n* = 28 per group) or ethanol diet (WT-EtOH and KO-EtOH groups; *n* = 28 per group), with equal numbers of female and male mice in each group. In our recently established ACLI model [2], the *Abcb4^−/−^* mouse model was combined with a standardized chronic plus binge ethanol feeding model (NIAAA model) by the National Institute on Alcohol Abuse and Alcoholism (NIAAA) [8]. In this model, mice were first fed with an ad libitum liquid diet (Rodent liquid diet, Lieber-DeCarli ‘82, Bio-Serv, Frenchtown, NJ) for five days. At the end of the 5th day, mice were either fed with a control liquid diet or ethanol diet (Rodent liquid diet, Lieber-DeCarli ‘82-Ethanol, 4% *v*/*v*, Bio-Serv) for a period of 10 days, followed by isocaloric maltose dextrin gavage (9 g/kg of body weight) or by an acute ethanol binge (4 g/kg of body weight), respectively. Mice were sacrificed 7–9 h after gavage, and plasma samples were prepared from whole blood collected from the inferior vena cava. Liver and intestine tissue samples were frozen in liquid nitrogen and stored at −80 °C. Bile collected from the gallbladder was stored at −20 °C. All experiments were performed in accordance with the relevant regulations and the Animal Care and Use Committee of Saarland University and were approved by Saarland University Animal Ethics Committee (TV 44/2015, date 22.09.2015 and TV 06/2019, date 04.02.2019).

### 4.2. Human Subjects

Blood samples were collected from volunteer patients with alcohol-associated liver disease (AALD) (*n* = 31, 24 men, 7 women) and healthy controls (*n* = 27, 14 men, 13 women) who provided their written informed consent. General clinical characteristics of the human study groups are given in Appendix A. The patients with alcoholic cirrhosis were recruited after hospital admission due to decompensation and admitted ongoing alcohol consumption (30–100 g EtOH daily) for at least one week prior to hospitalization. As the precise amount of EtOH intake could not be accurately defined, we measured carbohydrate-deficient transferrin (CDT) as a biochemical indicator for the relative amount of alcohol intake [34]. None of the healthy control volunteers had a history of acute or chronic liver disease. Healthy control subjects did not consume alcohol for at least three days before the day of blood collection. All healthy controls presented with normal body mass index (BMI) and glycated hemoglobin (HbA1c) levels (Appendix A). The study was approved by the Ethics Committee of Saarland Medical Board (approval number 271/11, date 16.10.2018).

### 4.3. Analysis of Plasma Biochemical Markers

Carbohydrate-deficient transferrin (CDT) was measured using the Siemens N Latex CDT kit (BNII; Siemens Healthcare Diagnostics, Erlangen, Germany). Alanine aminotransferase (ALT) and aspartate aminotransferase (AST) activities and ethanol (EtOH) levels were determined in plasma (Cobas 8000 c702 modular analyzer; Roche Diagnostics, Mannheim, Germany) in the Central Laboratory of Saarland University Medical Center.

### 4.4. Histology

Hepatic collagen accumulation was assessed by the collagen-specific amino acid hydroxyproline (Hyp) content and Sirius red histochemistry [63]. Inflammation in the liver was evaluated by hematoxylin-eosin (H&E) staining. For histopathological analysis, paraffin-embedded, 5 μm-thick formalin-fixed (4% *v*/*v*) liver sections were used [63]. Quantification of liver fibrosis and assessment of inflammation were performed using a histomorphometric semi-automatic system for image analysis (Leica microscope, equipped with Leica application suite software; Wetzlar, Germany) [2]. For each Sirius red-stained liver section, five randomly chosen microscopic areas were evaluated to calculate the percentage of collagenous areas. Photometric measurement of Hyp was performed by a protocol adapted from Jamall et al. [64].

#### Immunohistochemistry (IHC)

Paraffin-embedded liver tissue sections were stained for CD4 (Abcam; Cambridge, UK), CD11b (Abcam), CD45 (Biolegend; San Diego, CA, USA), and F4/80 (Abcam) to identify the inflammatory infiltrates. Deparaffinisation was performed in a descending alcohol series. Ten micromoles of citrate buffer (pH 6.0) in a microwave oven (600 W, 15 min) was used for antigen retrieval. Samples were then incubated for 20 min in 3% (*v*/*v*) H_2_O_2_ in PBS to block the endogenous peroxidase. After blocking with an avidin and biotin system (Dako; Glostrup, Denmark), sections were washed thrice with PBS, followed by overnight at 4 °C (for CD45 and CD4) and 1 h at 37 °C (for CD11b and F4/80) incubations with the primary antibody (1:100, 1:4000, 1:1000, and 1:400 dilutions in 2% (*w*/*v*) milk powder in PBS for CD45, CD4, CD11b, and F4/80, respectively). Afterward, sections were treated with secondary antibody (biotinylated-goat anti-rabbit, 1:100 diluted in 2% (*w*/*v*) milk powder in PBS) and ABC complex (Vectastain; Vector Laboratories, Burlingam, CA, USA). Nuclei staining was performed by counter-staining with Mayer’s hematoxylin as final step.

### 4.5. Determination of Plasma FGF15/FGF19 and FGF21 Levels

Plasma levels of FGF21 and FGF15 were measured in mice by the FGF-21 Quantikine ELISA kit (MF2100; R&D Systems, Minneapolis, MN, USA) and mouse FGF15 ELISA kit (RD-FGF15-Mu; Reddot Biotech, Kelowna, BC, Canada), respectively. For human plasma samples, the FGF-21 Quantikine ELISA kit (DF2100; R&D Systems) and FGF-19 Quantikine ELISA kit (DF1900; R&D Systems) were used. The experiments were carried out in duplicates, and average values were used for analysis.

### 4.6. RNA Isolation and Quantitative Real-Time PCR Analysis

Total RNA was isolated from frozen liver samples using the RNeasy Mini kit (Qiagen, Hilden, Germany). For ileum tissue and primary mouse hepatocyte cell culture experiments, peqGold Microspin Total RNA kit (VWR International, Erlangen, Germany) was used. Isolated total RNA was reverse-transcribed using the High-Capacity cDNA Reverse Transcription kit (Life Technologies, Carlsbad, CA, USA). The cDNA was amplified in 96-well plates using TaqMan^®^ Fast Universal PCR Master Mix (2×) (Applied Biosystems/Thermo Scientific, Foster City, CA, USA) on a Taqman 7500 Fast Real-Time PCR System. Ready-to-use TaqMan gene expression assays (Applied Biosystems/Thermo Scientific) were used for the evaluation of hepatic steady-state mRNA levels of Sterol regulatory element binding transcription factor 1 (*Srebf1/Srebp1c*, Mm00550338_m1), Fibroblast growth factor receptor 4 (*Fgfr4*, Mm01341852_m1), Fibroblast growth factor receptor 1 (*Fgfr1*, Mm00438930_m1), Cytochrome P450 family 7 subfamily A member 1 (*Cyp7a1*, Mm00484150_m1), Fibroblast growth factor 21 (*Fgf21*, Mm00840165_m1), Farnesoid X-activated receptor (*Fxr/Nr1h4*, Mm00436425_m1), β-Klotho (*Klb*, Mm00473122_m1), Mechanistic target of rapamycin (*Mtor*, Mm00444968_m1), Peroxisome proliferator activated receptor-α (*Ppar-a*, Mm00440939_m1), Small heterodimer partner (*Shp/Nr0b2*, Mm00442278_m1), Cytochrome P450 family 27 subfamily A member 1 (*Cyp27a1*, Mm00470430_m1), and Cytochrome P450 family 8 subfamily B member 1 (*Cyp8b1*, Mm00501637_s1) genes, as well as ileal mRNA levels of *Fxr*, MAM, and LDL receptor class A domain containing 1-Diet1 (*Malrd1/Diet1,* Cg04494263_m1), and Fibroblast growth factor 15 (*Fgf15*, Mm00433278_m1) genes. Relative changes in the expression of genes were evaluated using the ΔΔC_t_ algorithm, with Glyceraldehyde-3-phosphate dehydrogenase (*Gapdh*, Mm99999915_g1) as an internal control.

### 4.7. Western Blot Analysis

Mouse liver tissues (two randomly selected mice per group) were homogenized in lysis buffer (Tris-HCl 100 mmol/L, sodium dodecyl sulfate (SDS) 4%, glycine 20%) that contains complete protease inhibitors (cOmplete; Roche, Mannheim, Germany) and 1 mM of phenylmethylsulfonyl fluoride, and centrifuged at 16,000× *g* for 10 min. Isolated protein was quantified based on the method of Bradford [65], and 50 µg of protein was loaded in each well. Proteins were separated by 10% SDS-PAGE and transblotted onto nitrocellulose membranes (0.2 µm pore size, Schleicher and Schuell, Dassel, Germany). Blots were blocked in Tris-buffered saline containing 5% non-fat dry milk for 2 h at room temperature and incubated overnight at 4 °C with the following primary antibodies: anti-glyceraldehyde-3-phosphate dehydrogenase (GAPDH; MAB374, Millipore; Burlington, MA, USA), anti-CYP7A1 (ab65596, Abcam), anti-FGF21 (ab171941, Abcam), anti-FXR (#PA5-40755, Invitrogen), anti-mTOR (#2972, Cell Signaling Technology), and anti-phospho-mTOR–Ser2448 (#2971, Cell Signaling Technology). Blots were then incubated with respective secondary antibodies (anti-mouse #A5278 or anti-rabbit #A6154, Sigma, St. Louis, MO, USA) for 1 h at room temperature and were visualized by enhanced chemiluminescence (ECL, Amersham Biosciences; Buckinghamshire, UK). Quantification was performed using the Fusion SL gel documentation system (PEQLAB, Erlangen, Germany). All data are presented as intensity optical density (IOD).

### 4.8. Total Bile Acid Measurements in Mouse Plasma and Bile

Gallbladder bile was collected from a total of 37 mice (11 WT-Cont, 11 WT-EtOH, 9 KO-Cont, and 6 KO-EtOH). Total bile acid (TBA) concentrations were determined by an enzymatic colorimetric kit (BI 3863; Randox Laboratories, Wülfrath, Germany). The taurocholic acid standard was prepared from powdered taurocholic acid in methanol and stored at −20 °C. Bile acid standard solution was used in different dilution series for gallbladder (range 5–160 µM) and plasma samples (range 0.5–60 µM). Each sample was studied in duplicates, and average values were used for analysis. Two absorption measurements were carried out in a microplate reader at 405 nm at times t0 and t1 (i.e., two minutes later than t0). The difference in absorbance E(t1)-E(t0) is considered to be proportional to the bile acid concentration. The proportionality factor corresponded to the slope of the standard dilution curve.

### 4.9. Bile Acid Profiling in Human and Murine Plasma Samples

Bile acid profiling was performed using plasma samples of male mice and humans (randomly selected 3 samples/groups). An aliquot of human or murine plasma was incubated for one hour at 90 °C or for 2 h, respectively, using a strong alkaline hydrolysis in order to deconjugate the individual bile acids. After separation from the neutral sterols such as cholesterol and noncholesterol sterols by cyclohexane, the free bile acids were extracted after neutralization of the hydrophilic layer by diethyl ether. After evaporation of the solvents, the free bile acids were conjugated into methyl esters and trimethylsilyl ethers together with the internal standard, hyodeoxycholic acid. The plasma concentrations were quantified using gas chromatography–mass spectrometry–selected ion monitoring (GC-MS-SIM), as described previously [66]. In comparison to the bile acid analysis of human plasma samples (primary bile acids: cholic acid [CA] and chenodeoxycholic acid [CDCA], secondary bile acids: deoxycholic acid [DCA], lithocholic acid [LCA], and ursodeoxycholic acid [UDCA]) in murine plasma samples, we additionally determined α- and β-muricholic acid.

### 4.10. Enzymatic Tests for the Analysis of Serum Total Bile Acid and Cholesterol Levels in Humans

Serum total bile acid (TBA) concentrations were measured using the LT-SYS^®^ Gallensäuren test (LT-SYS Diagnostics; Berlin, Germany) with a Randox-Calibrator Cal 2351 and total cholesterol concentrations were measured by an enzymatic colorimetric determination assay using the Cholesterol Gen.2 (Cobas; Roche Diagnostics). Experiments were performed in the Central Laboratory of Saarland University Medical Center.

### 4.11. Quantification of Human and Murine Serum Concentrations of Total Cholesterol and Oxysterols Using Gas-Chromatography Analyses

Neutral sterols (cholesterol and noncholesterol sterols) and the oxysterols 7α- and 27-hydroxycholesterol (7α-OHC and 27-OHC, respectively) were extracted by cyclohexane in its free form after deconjugation of the fatty acid esterified sterols/oxysterols by alkaline hydrolysis. After conjugation into the corresponding trimethylsilyl ethers, serum total cholesterol was quantified by gas chromatography–flame ionization detection (GC-FID) using 5α-cholestane as an internal standard [67,68]. Serum concentrations of 7α-OHC and 27-OHC were quantified by the isotope dilution GC–mass spectrometry–selected ion monitoring (MS-SIM) methodology, using the corresponding deuterium-labeled oxysterols as internal standards [68].

### 4.12. Primary Mouse Hepatocyte Isolation, Culture, and Treatment

Primary mouse hepatocytes (HC) were isolated from C57BL/6J mice (8–15 week old) by the collagenase perfusion method [69]. Cell viability was assessed by trypan blue exclusion. Isolated primary HC were then seeded in 6-well collagen-I-coated (collagen from rat tail; Sigma-Aldrich, St. Louis, MO, USA) cell culture plates at a density of 3 × 10^5^ viable cells per well and incubated in 5% CO_2_ at 37 °C for four hours. After attachment of the cells by incubation with medium 1 (Williams’ medium E supplemented with 10% fetal bovine serum (FBS), 2 mM of L-glutamine, 1% penicillin/streptomycin, and 100 nM of dexamethasone), medium 1 was changed to medium 2 (medium 1 without FBS) for serum starvation and cell cycle synchronization. After 20 h, cells were treated with or without graded concentrations of recombinant human FGF21 (rhFGF21) (0.1, 0.25, 0.5, and 1 µg/mL; R&D Systems) in medium 3 (Williams’ medium E supplemented with 2 mM of L-glutamine and 1% penicillin/streptomycin) for 24 h. The cells were then harvested for RNA extraction and gene expression analysis. PBS was included as a control. Experiments were performed in triplicates, and each experiment was repeated three times.

### 4.13. Statistical Analysis

Statistical analysis was performed using SPSS v27.0 (IBM; Armonk, NY, USA) and GraphPad Prism 8 (GraphPad; San Diego, CA, USA). The normal distribution was first checked via the Kolmogorov–Smirnov test. The one-way ANOVA test was used for the normally distributed variables, and the Kruskal–Wallis test was used with data showing nonparametric distribution. A post hoc test with Bonferroni correction was performed when there were significant differences between the groups. A Student’s *t*-test was conducted for the two normally distributed variables as indicated. All results are shown in median, +max/−min. A *p*-value of <0.05 was considered statistically significant.

## Figures and Tables

**Figure 1 ijms-22-07898-f001:**
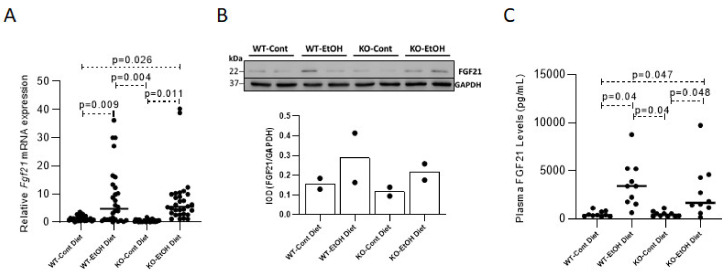
The effect of ethanol on FGF21 levels in liver and plasma. Relative quantification of Fgf21 mRNA levels in liver (*n* = 28 per group) (**A**). Western blot quantification of FGF21 protein in mouse liver by integrated optical density (IOD) that was normalized to GAPDH as a control (**B**) (*n* = 2 per group, each lane represents one mouse) and plasma FGF21 levels quantified by ELISA (**C**) (*n* = 10 per group). *p* < 0.05 was considered significant. Cont, control diet; EtOH, ethanol diet; KO, *Abcb4* knockout mice; WT, C57BL/6J wild-type mice.

**Figure 2 ijms-22-07898-f002:**
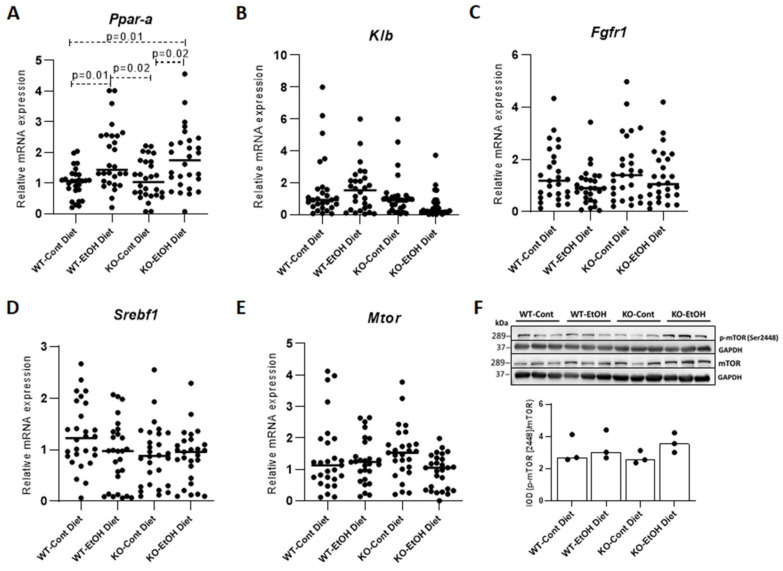
Relative quantification of hepatic mRNA levels. Transcriptional levels of *Ppar-a* (**A**), *Klb* (**B**), *Fgfr1* (**C**), *Srebf1* (**D**), and *Mtor* (**E**) were determined by qPCR in the liver (*n* = 28 per group). Western blot analysis of hepatic p-mTOR (Ser2448) and total mTOR protein expressions (*n* = 3 per group, each lane represents one mouse) and quantitative analysis of p-mTOR/mTOR by IOD normalized to GAPDH as a control (**F**). Cont, control diet; EtOH, ethanol diet; IOD, integrated optical density; KO, *Abcb4* knockout mice; p, phospho; WT, C57BL/6J wild-type mice. *p* < 0.05 was considered significant.

**Figure 3 ijms-22-07898-f003:**
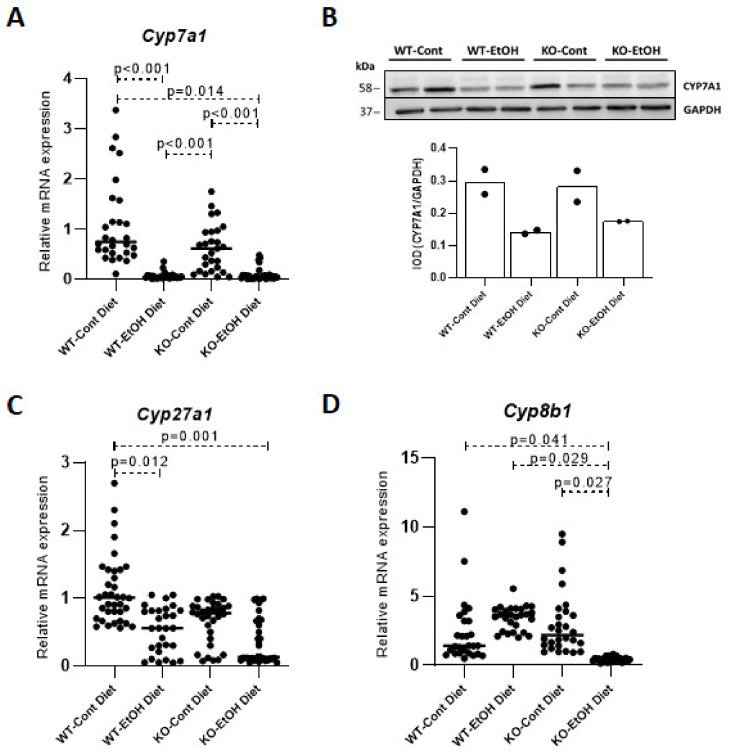
Relative quantification of gene expression using qPCR in liver. Hepatic mRNA levels of *Cyp7a1* gene (**A**), Western blot analysis of CYP7A1 protein in mouse liver by IOD that was normalized to GAPDH as a control (*n* = 2, each lane represents one mouse) (**B**) and hepatic mRNA levels of bile acid synthesis genes *Cyp27a1* (**C**) and *Cyp8b1* (**D**) (*n* = 28 per group). Cont, control diet; EtOH, ethanol diet; IOD, integrated optical density; KO, *Abcb4* knockout mice; WT, C57BL/6J wild-type mice. *p* < 0.05 was considered significant.

**Figure 4 ijms-22-07898-f004:**
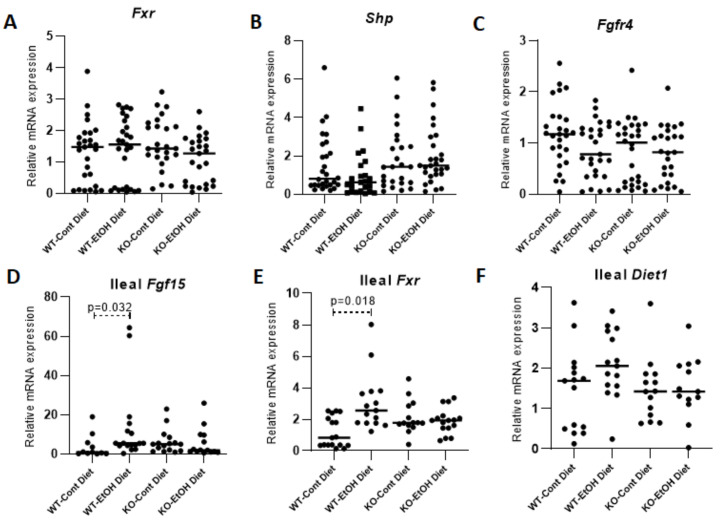
Relative quantification of mRNA levels of hepatic and ileal genes that play a role in transcriptional regulation of *Cyp7a1*. Transcriptional levels of *Fxr* (**A**), *Shp* (**B**), and *Fgfr4* (**C**) in the liver (*n* = 28 per group), and *Fgf15* (**D**), *Fxr* (**E**), and *Diet1* (**F**) in the ileum were determined (*n* = 15 per group). Cont, control diet; EtOH, ethanol diet; KO, *Abcb4* knockout mice; WT, C57BL/6J wild-type mice. *p* < 0.05 was considered significant.

**Figure 5 ijms-22-07898-f005:**
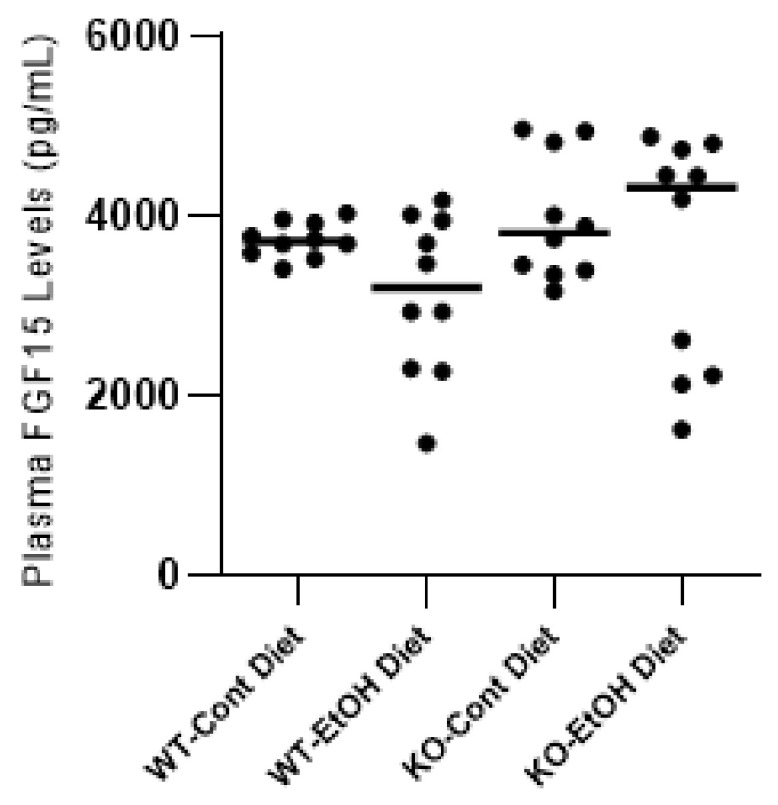
Plasma FGF15 levels in mice quantified by ELISA (*n* = 10 per group). *p* < 0.05 was considered significant. Cont, control diet; EtOH, ethanol diet; KO, *Abcb4* knockout mice; WT, C57BL/6J wild-type mice.

**Figure 6 ijms-22-07898-f006:**
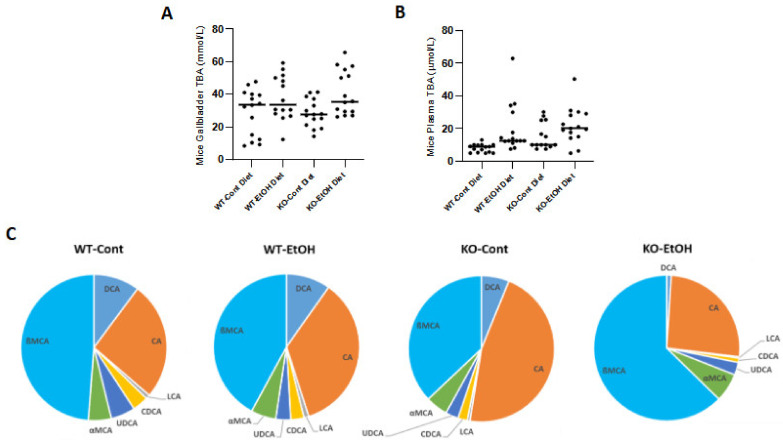
Total bile acid levels in male mouse gallbladder bile (**A**) and plasma (**B**) (*n* = 14 per group). Plasma individual bile acids were identified by GC-MS-SIM (*n* = 3 per group) and are shown as the percentage of total bile acid pool (**C**). Secondary bile acids comprise DCA, LCA, and UDCA. CA, cholic acid; CDCA, chenodeoxycholic acid; Cont, control diet; DCA, deoxycholic acid; EtOH, ethanol diet; KO, *Abcb4* knockout mice; LCA, lithocholic acid; UDCA, ursodeoxycholic acid; WT, C57BL/6J wild-type mice; βMCA, β-muricholic acid; αMCA, α-muricholic acid.

**Figure 7 ijms-22-07898-f007:**
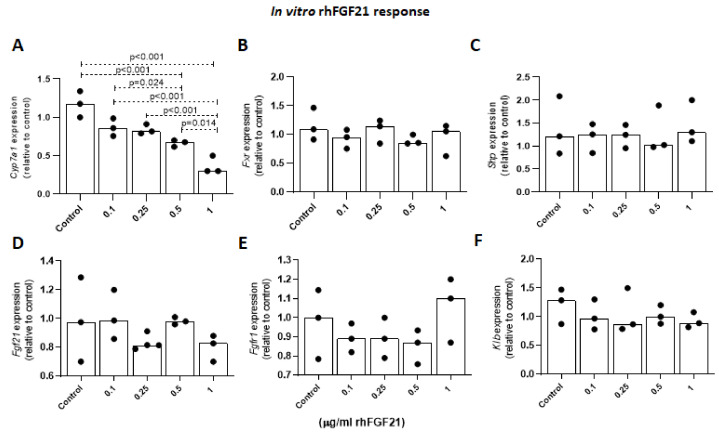
The effects of rhFGF21 (µg/mL) on primary mouse hepatocytes. Transcriptional levels of *Cyp7a1* (**A**), *Fxr* (**B**), *Shp* (**C**), *Fgf21* (**D**), *Fgfr1* (**E**), and *Klb* (**F**) genes in primary mouse hepatocytes not treated with rhFGF21 (control) or cells treated with rhFG21 at different doses (0.1–1 µg/mL) for 24 h. All data represent x-fold change as compared to control hepatocytes (*n* = 3 independent primary mouse hepatocyte cultures). *p* < 0.05 was considered significant.

**Figure 8 ijms-22-07898-f008:**
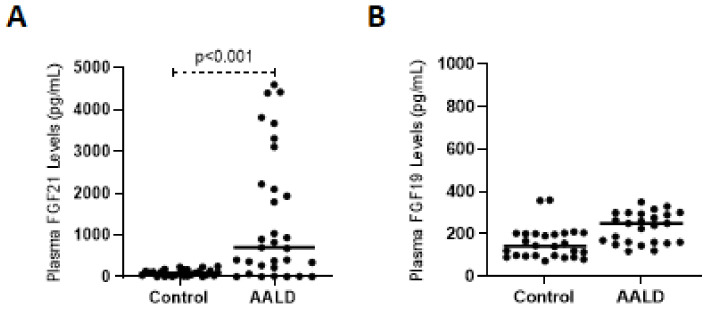
Box-and-whisker plots showing plasma FGF21 (**A**) and FGF19 (**B**) levels in human plasma samples quantified by ELISA (AALD, *n* = 31, controls, *n* = 27). AALD, alcohol-associated liver disease. *p* < 0.05 was considered significant.

**Figure 9 ijms-22-07898-f009:**
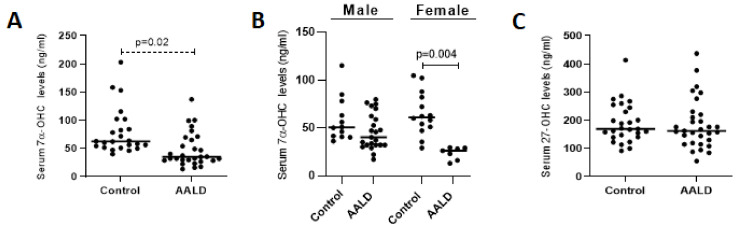
Serum concentrations of 7α-OHC (**A**), 7α-OHC stratified for sex (**B**), and 27-OHC (**C**), as measured by GC-MS-SIM. (AALD, *n* = 31, 24 men, 7 women, and controls, *n* = 27, 14 men, 13 women). AALD, alcohol-associated liver disease. *p* < 0.05 was considered significant.

**Figure 10 ijms-22-07898-f010:**
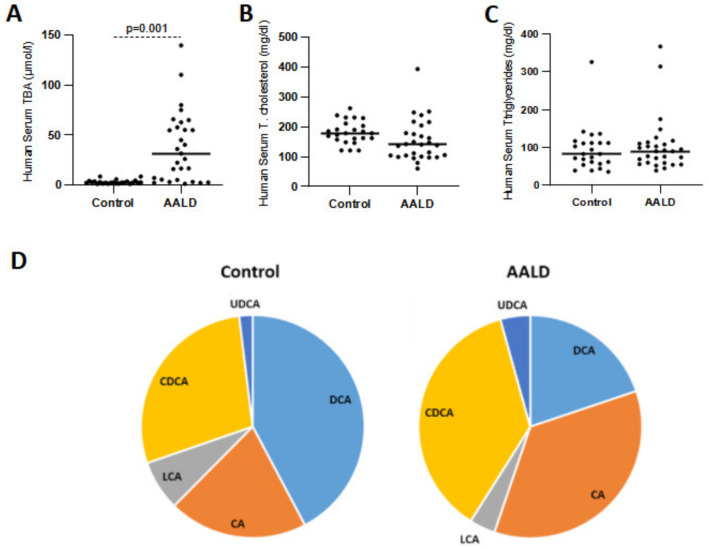
Serum total bile acid (**A**), cholesterol (**B**), and triglyceride (**C**) concentrations in humans. Individual bile acids were identified by GC-MS-SIM (men, *n* = 3 per groups) and are shown as the percentage of total bile acid pool (**D**). AALD, alcohol-associated liver disease, CA, cholic acid; CDCA, chenodeoxycholic acid; Cont, control diet; DCA, deoxycholic acid; LCA, lithocholic acid; UDCA, ursodeoxycholic acid.

**Figure 11 ijms-22-07898-f011:**
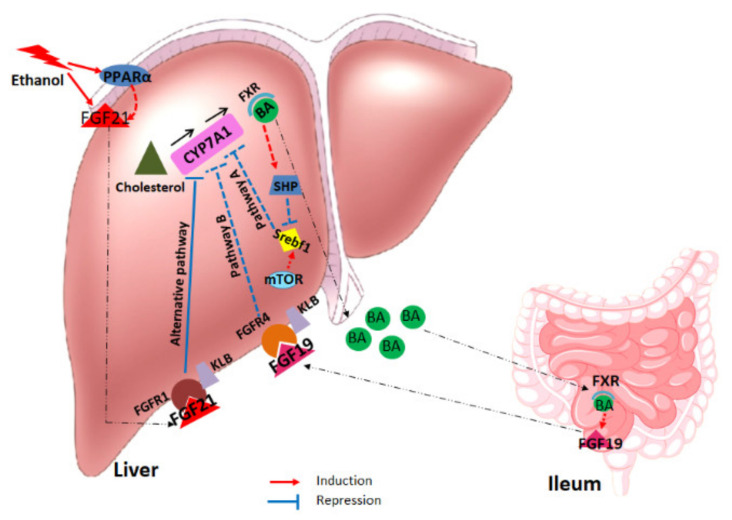
Suggested alternative pathway for bile acid synthesis regulation upon ethanol challenge. Dashed lines for induction and repression represent literature knowledge (Pathways A and B), and straight lines indicate the results of the current study.

**Table 1 ijms-22-07898-t001:** Percentage of total bile acid pool in mice gallbladder.

% (mean ± SD)	CA	CDCA	DCA	LCA	UDCA	ßMCA	αMCA	Primary BA	Secondary BA
Wt-Cont	26.3 ± 2.1	3.5 ± 1.7	10.3 ± 3.3	0.9 ± 0.1	5.4 ± 1.8	48.5 ± 8.7	5.1 ± 1.2	83.4%	16.6%
Wt-EtOH	35.3 ± 2.3	2.9 ± 0.6	9.8 ± 3.1	1.1 ± 0.2	3.3 ± 0.9	42.1 ± 0.7	5.5 ± 1.5	85.8%	14.2%
KO-Cont	46.2 ± 4.7	2.1 ± 1.2	6.5 ± 1.6	0.6 ± 0.2	2.8 ± 1.0	36.9 ± 3.1	4.9 ± 1.0	90.1%	9.9%
KO-EtOH	25.9 ± 1.3	1.0 ± 0.2	1.0 ± 0.2	0.3 ± 0.1	2.8 ± 0.2	62.7 ± 2.2	6.3 ± 1.1	95.9%	4.1%

Percentages are shown as % mean ± SD. Secondary bile acids comprise DCA, LCA, and UDCA. CA, cholic acid; CDCA, chenodeoxycholic acid; Cont, control diet; DCA, deoxycholic acid; EtOH, ethanol diet; KO, *Abcb4* knockout mice; LCA, lithocholic acid; UDCA, ursodeoxycholic acid; WT, C57BL/6J wild-type mice; βMCA, β-muricholic acid; αMCA, α-muricholic acid.

**Table 2 ijms-22-07898-t002:** Percentage of total bile acid pool in human serum.

% (mean ± SD)	CA	CDCA	DCA	LCA	UDCA	Primary BA	Secondary BA
Control	20.6 ± 5.3	29.1 ± 2.1	40.8 ± 5.8	7.5 ± 1.8	2.0 ± 0.4	49.7%	50.3%
AALD	35.5 ± 7.6	36.7 ± 3.6	19.8 ± 4.3	3.7 ± 1.3	4.3 ± 1.3	72.2%	27.8%

Percentages are shown as % mean ± SD. AALD, alcohol-associated liver disease, CA, cholic acid; CDCA, chenodeoxycholic acid; Cont, control diet; DCA, deoxycholic acid; LCA, lithocholic acid; UDCA, ursodeoxycholic acid.

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
