# Peer review of "Fibroblast Growth Factor 21 Response in a Preclinical Alcohol Model of Acute-on-Chronic Liver Injury"

_ijms, 2021, doi:10.3390/ijms22157898_

Round 1

Reviewer 1 Report

Fibroblast growth factor (FGF) 21 has been shown to play a potential role in bile acid metabolism, but the mechanism has been undetermined. The authors investigated the FGF21 response in an ethanol-induced acute-on-chronic liver injury model in Abcb4-/- mice with deficiency of the hepatobiliary phospholipid transporter. They demonstrated simultaneous upregulation of FGF21 and downregulation of Cyp7a1 expression upon chronic plus binge alcohol feeding, whereas plasma FGF15 and hepatic Shp and Fxr levels were unchanged. In hepatocyte in primary culture, FGF21 suppressed Cyp7a1 expression. These results suggest the presence of a direct regulatory mechanism of FGF21 on bile acid homeostasis through inhibition of CYP7A1 in an ethanol-induced acute-on-chronic liver injury model.

This study is important and seems to be performed in a well-conducted manner. Careless mistakes were not found. Experiments are extensive and the manuscript is carefully prepared. I have no comments to request additional experiments and changes.

Author Response

We thank the Reviewer for the time and effort invested and all positive comments on the manuscript.

Reviewer 2 Report

This paper by Christidis et al. shows that EtOH consumption results in elevated FGF21 plasma concentrations which affects bile acid biosynthesis. The strength of this study is that authors show effect of EtOH consumption on FGF21 and bile acid metabolism in a pre-clinical model of alcoholism and in patients with high EtOH consumption. However, authors do not emphasize the genotype dependent differences in either effect or effect magnitude of EtOH consumption. This paper can be published after following concerns have been addressed:

The authors could add a figure with the model of molecular interactions and regulatory circuits that connect FGF21 and CYP7A1 that was tested in this study.

Number of assayed samples should be added in each figure with measurements of gene expression, plasma concentrations of bile acids and FGF, gallbladder concentrations of bile acids etc. … (Figures 1-6, Figure S1, S2C-D).

Individual data points should be plotted in all figures to evaluate the distribution of the measurements. Min/max intervals are asymmetrical in many graphs suggesting that outliers may drive the differences in the means.

The authors measured mRNA levels of Ppar-a, Mtor, Srebf1, Fgfr1, Klb, Fxr, Shp, Fgfr4, Fgf15, and Diet1 in mouse primary hepatocytes, livers and intestines. Based on these observation authors make conclusions about involvement of different pathways and regulators in expression of FGF21. These conclusions can be valid only if it is known that mRNA and protein levels correlate for all these genes and that post-translational modifications do not modify activities of gene products. This is especially true for mTOR – its activity depends on the phosphorylation state of specific amino acids and mTOR phosphorylation levels were not measured in this study. The authors should indicate for each one of the genes listed above whether their mRNA and protein levels correlate and whether post-translational modifications affect activity/function of each protein, and take that into account when discussing whether FXR/SHP or FGF receptors or PPAR-A and mTOR regulate FGF21 and CYP7A1 gene expression. The authors cannot make conclusions about role of regulatory proteins and pathways if mRNA and protein levels of the genes listed above do not correlate and/or post-translational modifications play a role in the function of proteins encoded by the genes. In that case, authors should either measure relevant protein and phosphorylation levels or refocus the manuscript to translational aspects of their pre-clinical model to human studies. This is especially important for the validity of statements in lines 550-555.

Indicate and discuss whether total plasma concentration of FGF-15 reflects its concentration in portal circulation.

The magnitude and direction of the effect of EtOH diet did not depend on the genotype for FGF21 expression, Ppar-a and Cyp7a1. Response to EtOH diet was genotype dependent for Cyp27a1, Cyp8b1, ileal Fgf15 and Fxr expression and concentration changes of some bile acids. In this context authors again discuss effects of bile acids on Fxr expression at the mRNA level without taking into account possible regulation of FXR expression at the level of translation and post-translational modifications.

Bar graphs with individual data points are more suitable for Figure 7.

Minor comments:

Lines 43-46: Add details about human vs. animal studies when describing different models of EtOH induced liver injury and its effects on health.

Lines 49-52: Describe function of the Abcb4 gene.

Lines 74-77: Rewrite the sentence: Although the studies of the effects of alcohol on Cyp7a1 gene expression report controversial results [19, 23], it was suggested recently that CYP7A1 contributes to the protection against alcohol-associated steatohepatitis [24].

Lines 96-98: Rewrite the sentence: To translate our findings in mice to humans, we also compared the effect of alcohol intake on FGF21, FGF19 and bile acid metabolism in human subjects with alcohol-associated liver cirrhosis and healthy controls.

Author Response

We would like to thank for the valuable and insightful feedbacks of the reviewers to improve our manuscript entitled as „Fibroblast growth factor 21 response in a preclinical alcohol model of acute-on-chronic liver injury“. We have been able to revise the manuscript based on the suggestions provided by the reviewers and highlighted the changes throughout the manuscript. Please find below the point-by-point response to the reviewers’ comments and concerns:

This paper by Christidis et al. shows that EtOH consumption results in elevated FGF21

plasma concentrations which affects bile acid biosynthesis. The strength of this study is

that authors show effect of EtOH consumption on FGF21 and bile acid metabolism in a

preclinical model of alcoholism and in patients with high EtOH consumption. However,

authors do not emphasize the genotype dependent differences in either effect or effect

magnitude of EtOH consumption. This paper can be published after following concernes

have been addressed:

The authors could add a figure with the model of molecular interactions and regulatory

circuits that connect FGF21 and CYP7A1 that was tested in this study.

  • The figure is now provided in the discussion section (Figure 11, page 27) as suggested by the reviewer.

Number of assayed samples should be added in each figure with measurements of

gene expression, plasma concentrations of bile acids and FGF, gallbladder

concentrations of bile acids etc. … (Figures 1-6, Figure S1, S2C-D).

  • Number of assayed samples is given in figure legend of each figure.

Individual data points should be plotted in all figures to evaluate the distribution of the

measurements. Min/max intervals are asymmetrical in many graphs suggesting that

outliers may drive the differences in the means.

  • Graphs are re-prepared and replaced according to the suggestion of the reviewer.

The authors measured mRNA levels of Ppar-a, Mtor, Srebf1, Fgfr1, Klb, Fxr, Shp, Fgfr4,

Fgf15, and Diet1 in mouse primary hepatocytes, livers and intestines. Based on these

observation authors make conclusions about involvement of different pathways and

regulators in expression of FGF21. These conclusions can be valid only if it is known

that mRNA and protein levels correlate for all these genes and that post-translational

modifications do not modify activities of gene products. This is especially true for mTOR

– its activity depends on the phosphorylation state of specific amino acids and mTOR

phosphorylation levels were not measured in this study. The authors should indicate for

each one of the genes listed above whether their mRNA and protein levels correlate

and whether post-translational modifications affect activity/function of each protein, and

take that into account when discussing whether FXR/SHP or FGF receptors or PPAR-A

and mTOR regulate FGF21 and CYP7A1 gene expression. The authors cannot make

conclusions about role of regulatory proteins and pathways if mRNA and protein levels

of the genes listed above do not correlate and/or post-translational modifications play a

role in the function of proteins encoded by the genes. In that case, authors should either

measure relevant protein and phosphorylation levels or refocus the manuscript to

translational aspects of their pre-clinical model to human studies. This is especially

important for the validity of statements in lines 550-555.

  • We agree with the reviewer`s suggestion on the correlation of mRNA/protein levels correlation and post translational modifications that may affect the function of the gene products, In an attempt to investigate these correlations, we did Western blot analysis for most but not all of these gene products due to limited time schedule for revisions, Moreover, some of the antibodies ordered to be used in western analysis did not work during these efforts. However, we were able to determine and provide protein levels for FGF21 (Figure 1B), mTOR (Figure 2F), CYP7A1 (Figure 3B) and FXR (supplementary figure 1), which we think are the most critical gene products in our study. Therefore, correlation between mRNA and protein levels of these gene products are now revealed by Western blots. Moreover, as also suggested by the reviewer, the ratio of phospho-mTOR to total mTOR was found to be similar between the groups, suggesting that no difference was present between groups in terms of post-translational modifications. The results were also given (page 7; lines 360-364, page 8, Figure 2F) and discussed in the manuscript where appropriate (page 17, lines 771-774). Reference 55 is included accordingly.

Indicate and discuss whether total plasma concentration of FGF-15 reflects its concentration in portal circulation.

  • This issue was indicated and discussed in the manuscript for both humans and mice (page 10, lines 531-537 and page 14, lines 643-646). References 46, 48, 49 are included accordingly.

The magnitude and direction of the effect of EtOH diet did not depend on the genotype for FGF21 expression, Ppar-a and Cyp7a1. Response to EtOH diet was genotype dependent for Cyp27a1, Cyp8b1, ileal Fgf15 and Fxr expression and concentration changes of some bile acids. In this context authors again discuss effects of bile acids on Fxr expression at the mRNA level without taking into account possible regulation of FXR expression at the level of translation and post-translational modifications.

  • As discussed above, although post-translational modifications has not been checked for FXR, mRNA/protein correlation was confirmed by western blot and included as a supplementary figure (page 24, Supplementary figure 1) in the manuscript. However, one limitation was that due to limited amount of ileum samples, Western Blots were performed from liver samples.

Bar graphs with individual data points are more suitable for Figure 7.

  • Figure 7 was revised accordingly (page 13).

Minor comments:

Lines 43-46: Add details about human vs. animal studies when describing different

models of EtOH induced liver injury and its effects on health.

  • Details are given as requested by the reviewer (page 2, lines 49-59).

Lines 49-52: Describe function of the Abcb4 gene.

  • Function of the Abcb4 gene is now defined (page 2, lines 64-67).

Lines 74-77: Rewrite the sentence: Although the studies of the effects of alcohol on

Cyp7a1 gene expression report controversial results [19, 23], it was suggested recently

that CYP7A1 contributes to the protection against alcohol-associated steatohepatitis

[24].

       - Sentence is re-written (page 2, lines 92-95).

Lines 96-98: Rewrite the sentence: To translate our findings in mice to humans, we

also compared the effect of alcohol intake on FGF21, FGF19 and bile acid

metabolism in human subjects with alcohol-associated liver cirrhosis and healthy

controls.

  • Sentence is re-written (page 3, lines 126-130).

Round 2

Reviewer 2 Report

The authors addressed all my comments. Please double check that the text refers to correct panels in figures. I noticed that text refers to Figure 3C (line 372) but it should be Figure 3D.

Author Response

We would like to thank the reviewer for her/his comment. We now double checked the figure names throughout the text and did the following corrections:

  • In page 8, line 461: „Fig. 3C“ corrected as „Fig. 3D“.
  • In page 11, line 567: „Fig 6D“ corrected as „Fig. 6C“.
  • In page 13, line 618: „Supplementary Fig. 3“ corrected as „Supplementary Fig. 4“.
  • In page 16, line 753: „figure 9C“ corrected to „Fig. 9B“.
